# A Critical Evaluation of Terpenoid Signaling at Cannabinoid CB1 Receptors in a Neuronal Model

**DOI:** 10.3390/molecules27175655

**Published:** 2022-09-02

**Authors:** Michaela Dvorakova, Sierra Wilson, Wesley Corey, Jenna Billingsley, Anaëlle Zimmowitch, Joye Tracey, Alex Straiker, Ken Mackie

**Affiliations:** Gill Center for Molecular Bioscience, Department of Psychological and Brain Sciences, Indiana University, Bloomington, IN 47405, USA

**Keywords:** G protein-coupled receptor, cannabinoid, cannabis, CB1, terpene, terpenoid

## Abstract

In addition to phytocannabinoids, cannabis contains terpenoids that are claimed to have a myriad of effects on the body. We tested a panel of five common cannabis terpenoids, myrcene, linalool, limonene, α-pinene and nerolidol, in two neuronal models, autaptic hippocampal neurons and dorsal root ganglion (DRG) neurons. Autaptic neurons express a form of cannabinoid CB1 receptor-dependent retrograde plasticity while DRGs express a variety of transient receptor potential (TRP) channels. Most terpenoids had little or no effect on neuronal cannabinoid signaling. The exception was nerolidol, which inhibited endocannabinoid signaling. Notably, this is not via inhibition of CB1 receptors but by inhibiting some aspect of 2-arachidonoylglycerol (2-AG) production/delivery; the mechanism does not involve reducing the activity of the 2-AG-synthesizing diacylglycerol lipases (DAGLs). Nerolidol was also the only terpenoid that activated a sustained calcium response in a small (7%) subpopulation of DRGs. In summary, we found that only one of five terpenoids tested had notable effects on cannabinoid signaling in two neuronal models. Our results suggest that a few terpenoids may indeed interact with some components of the cannabinoid signaling system and may therefore offer interesting insights upon further study.

## 1. Introduction

Cannabis has a long history of use among humans, stretching back thousands of years, but is experiencing a surge in use coincident with an ongoing shift in its legal status. Most attention has been paid to the two phytocannabinoids that are present in the greatest quantity—Δ^9^-tetrahydrocannabinol (Δ^9^-THC) and cannabidiol. Δ^9^-THC primarily exerts its psychoactive effects via cannabinoid CB1 receptors [1] that are widely distributed in the brain [2] and that regulate important physiological systems such as pain, mood, movement and memory (reviewed in [3]).

There is a rising interest, both in the 100+ related phytocannabinoids found in cannabis, as well as terpenoids, a class of compounds swept up by the same wave. Phytocannabinoids and terpenoids are being widely incorporated into products and marketed for their claimed health benefits [4]. Terpenoids are variants of terpenes, a large class of organic compounds that confer strain-specific odors to cannabis plants, though they likely also have other properties. Several cannabis-derived terpenoids have attracted the interest of companies and consumers; these terpenoids have been ascribed various effects, often proposed to be synergistic with phytocannabinoids (reviewed in [4,5]). Recent studies have tested the interaction of some of them with the cannabinoid signaling system in heterologous expression systems and found no effects on CB1 signaling [6,7]. However, the pharmacological properties of terpenoids have not thus far been examined in a neuronal model. We therefore examined a panel of the more widely promoted terpenoids including (with putative effect in parentheses): myrcene (altered THC-psychoactivity), linalool (anxiolysis), limonene (mood), α-pinene (memory), and nerolidol (sedation). These terpenoids were tested for their interaction with endogenous cannabinoid signaling in two neuronal models, autaptic hippocampal neurons [8] and dorsal root ganglion (DRG) neurons [9]. This represents a companion to a recent study of understudied phytocannabinoids, using the same models [9]. Autaptic hippocampal neurons are cultured pyramidal neurons from the hippocampus that are cultured on ‘islands’ such that some number of them are solitary, forming autaptic synapses onto themselves. Over the past 20 years, we have characterized the cannabinoid signaling in these autaptic neurons since they express an intact endogenous cannabinoid signaling system. This includes presynaptic CB1 receptors, postsynaptic enzymes that synthesize the endocannabinoid 2-arachidonoylglycerol (2-AG) and enzymatic machinery to metabolize 2-AG [10,11,12]. They also exhibit several forms of CB1-mediated neuronal plasticity including depolarization-induced suppression of excitation, a form of retrograde inhibition that we have used extensively for studies of cannabinoid neuropharmacology (e.g., [8,9]). DRGs are first-order sensory neurons the cell bodies of which reside in ganglia adjoining the spinal column. They are essential for transmitting sensory—including nociceptive—stimuli, and are known to express various transient receptor potential (TRP) channels, several of which are reportedly activated by endogenous and plant cannabinoids [13,14]. These systems afford not just the opportunity to test terpenoid interactions with CB1 and TRP-family receptors but also their interactions with other components of endocannabinoid signaling.

## 2. Results

### 2.1. Myrcene, Linalool, Limonene and α-Pinene Do Not Alter CB1 Signaling in Autaptic Hippocampal Neurons

#### 2.1.1. Myrcene

Myrcene is a monoterpene (Figure 1) that is popularly supposed to be the active ingredient underlying the “mango effect” reported by some cannabis users whereby eating a mango before smoking cannabis enhances the high. In principle, a compound such as myrcene might enhance cannabinoid signaling by directly activating cannabinoid receptors, positive allosteric modulation, enhanced synthesis of endocannabinoids or inhibition of their metabolism. Myrcene has been reported to have an antihypernociceptive effect in a mouse model of neuropathic pain [15]. Recently myrcene has been reported to activate TRPV1 channels, albeit at concentrations in excess of 100 µM, which are unlikely to be encountered by consumers of oral or inhaled terpenoids (see discussion) [16]. We tested the impact of myrcene on CB1 endocannabinoid signaling in autaptic hippocampal neurons using whole cell patch clamp recording. We found that myrcene has no significant effect on cannabinoid signaling at 1 μM, a concentration that was chosen for myrcene and other terpenoids tested because it represents a likely ceiling physiological concentration to be encountered by consumers (see discussion for more details on this). At 1 µM myrcene did not alter EPSC amplitudes (Figure 2A, EPSC charge relative to baseline (1.0 = no inhibition) after myrcene (1 µM): 1.04 ± 0.03, *n* = 8, *p* = 0.225 by one-sample *t*-test vs. baseline (1.0)), indicating that myrcene does not directly alter neurotransmission in this model system. Similarly, myrcene did not alter depolarization-induced suppression of excitation (DSE), a form of endogenous 2-AG and CB1-mediated retrograde signaling present in autaptic hippocampal neurons. As described in methods, successively longer depolarizations (50 ms, 100 ms, 300 ms, 500 ms, 1 sec, 3 sec, 10 sec) result in greater inhibition of EPSCs, yielding a ‘depolarization dose-response’ curve. A potentiator of cannabinoid signaling would be expected to shift this curve to the left, as we have seen for a positive allosteric modulator [17], while an inhibitor of cannabinoid signaling would be expected to shift the curve to the right as we have reported, for example, for negative allosteric modulators [18]. We find that myrcene does not affect the DSE response curve at 1 μM (Figure 2B, *n* = 6, *p* = 0.688 via 2-way ANOVA; F_(6,60)_ = 0.652).

#### 2.1.2. Linalool

Linalool is a monoterpene with a structure similar to that of myrcene (Figure 1) and anxiolytic properties have been ascribed to this compound [19,20]. Linalool is more commonly associated with lavender extracts but has recently been proposed to interact with the cannabinoid signaling system [21]. Testing linalool in our model at 1 μM, we find that linalool does slightly increase EPSC sizes (Figure 2C, EPSC charge relative to baseline (1.0 = no inhibition) after linalool (1 μM): 1.10 ± 0.03, *n* = 5; *p* = 0.039 by one-sample *t*-test vs. baseline (1.0)). However, we did not see a significant alteration in DSE responses (Figure 2D, *n* = 6, *p* = 0.366 via 2-way ANOVA; F_(6,60)_ = 1.113).

#### 2.1.3. Limonene

Limonene is a cyclic monoterpene (Figure 1) associated with citrus rind that is promoted by cannabis vendors as being anxiolytic and anti-depressant and beneficial for memory [22,23,24]. Limonene did not increase EPSCs in these neurons (Figure 2E, EPSC charge relative to baseline (1.0 = no inhibition) after limonene (1 μM): 1.03 ± 0.03, *n* = 6, *p* = 0.10 by one-sample *t*-test vs. baseline (1.0)). Limonene had no effect on cannabinoid DSE signaling (Figure 2F, *n* = 7, *p* = 0.164, via 2-way ANOVA; F_(6,72)_ = 1.584).

#### 2.1.4. α-Pinene

α-Pinene is a monoterpene (Figure 1) that has been proposed to enhance memory [25] and to act against inflammation [26]. α-Pinene is found in conifers and is often mentioned in connection with ‘forest bathing’ that supposedly has a beneficial effect on human health [27]. α-Pinene did not alter neurotransmission (Figure 2G, EPSC charge relative to baseline (1.0 = no inhibition) after α-pinene (1 μM): 1.04 ± 0.03, *n* = 9, *p* = 0.31 by one-sample *t*-test vs. baseline (1.0)). α-Pinene also had no effect on CB_1_ signaling (Figure 2H, *n* = 7, *p* = 0.058 via 2-way ANOVA; F_(6,72)_ = 2.151).

The ED50 and maximal inhibition values are shown in Table 1.

### 2.2. Nerolidol Inhibits Cannabinoid Signaling by Altering Endocannabinoid Production

Nerolidol is a sesquiterpene (Figure 1) that is described as having sedative properties [28]. It proved to have the most interesting profile of the terpenoids tested here. Nerolidol did not directly alter neurotransmission when applied (Figure 3A, EPSC charge relative to baseline (1.0 = no inhibition) after nerolidol (1 μM): 0.983 ± 0.016, *n* = 15; *p* = 0.319 by one-sample *t*-test vs. 1.0), but did have an inhibitory effect on DSE responses, suppressing the maximal DSE elicited at 1 μM, but not at 100 nM (Figure 3B, 1 μM: *n* = 7, *p* = 0.0006 at 3 s and *p* < 0.0001 at 10 s depolarizations, via 2-way ANOVA with Bonferroni post hoc test; F_(6,72)_ = 9.418, *p* < 0.0001; 100 nM: *n* = 9, *p* = 0.535 via 2-way ANOVA, F_(6,96)_ = 0.850).

Since DSE signaling is a function of both post-synaptic 2-AG production and presynaptic CB1 signaling, there are two sites of action where inhibition of DSE might occur. To further explore a possible presynaptic mechanism of action, we tested the effect of nerolidol on responses to bath-application of the endogenous cannabinoid 2-AG. If the effect of nerolidol involved an inhibition of CB1 activity/signaling, then a similar inhibition of bath-applied 2-AG should be seen. We did not, however, see an inhibition of 2-AG responses over a range of 2-AG concentrations that would be achieved in autaptic synapses following DSE, suggesting that the effect of nerolidol on DSE involves postsynaptic reduction of 2-AG production or transport (Figure 3C; 2-AG: *n* = 9, 2-AG + Nerolidol: *n* = 5, *p* = 0.756 via 2-way ANOVA with Bonferroni post hoc test; (F_(4,44)_ = 0.472)).

If nerolidol acts post-synaptically, then a likely target is the diacylglycerol lipases, enzymes that produce 2-AG [29]. We have previously shown that DAGLα and DAGLβ act cooperatively to produce 2-AG in autaptic hippocampal neurons [30]. We therefore examined the activity of nerolidol at DAGLα and DAGLβ using two approaches. In one set of experiments, we made use of an Enzchek lipase assay. The Enzchek probe is a non-fluorescent substrate for lipases that produces a green fluorescent product [31]. In CHO cells that overexpress DAGLα nerolidol (3 μM) had no effect on lipase activity (Figure 4A,B) (Figure 4B, nerolidol (3 μM) in DAGLα-CHO (lipase activity (increased fluorescence/min/ug protein ± SEM): 5.2 ± 1.25; nerolidol: 5.7 ± 1.6; *n* = 3; NS unpaired *t*-test, *p* = 0.80).

We additionally tested DAGLβ activity in brain tissue using HT-01, a probe developed to characterize DAGLβ activity [32]. We found that while the DAGLβ blocker KT109 competes with HT-01 for binding at DAGLβ, nerolidol does not (Figure 5, nerolidol (5 μM) relative to vehicle (1.0 = no effect): 0.98 ± 0.04; *n* = 2, NS by 1-tailed *t*-test vs. 1.0, *p* = 0.70). Thus, nerolidol does not appear to inhibit either DAGL.

### 2.3. Myrcene, Nerolidol, Linalool, Pinene and Limonene Do Not Consistently Activate TRP Channels in Dorsal Root Ganglion Neurons

Transient receptor potential (TRP) receptors are non-selective ion channels that have various physiological roles including regulation of pain [33]. Various cannabinoids are reported to modulate TRP channels, but there is little research on the effects of terpenoids [34,35,36]. One study reported that myrcene activated TRPV1 at high concentrations (5–150 μM) [16]. Another study that examined terpenoid interactions with TRP receptors reported no effect [37].

We tested the terpenoids in rat dorsal root ganglion neurons (DRGs), a neuronal model that natively express an array of TRP channels including TRPV1, TRPV3, TRPV4, TRPM8 and TRPA1 [38,39]. One of the challenges of working with DRGs is that they have non-uniform TRP expression, with subpopulations that differentially express the TRP receptor subtypes [40,41]. We therefore used a population approach, with a fluorescent probe that simultaneously measured changes in intracellular calcium in populations of DRGs in response to terpenoids in real-time. In a given experiment we used the TRPV1 agonist capsaicin (1 μM) to confirm the responsiveness of DRGs; capsaicin activated ~40% of cells, which is consistent with the literature [35].

Of the terpenoids, nerolidol (1 μM) was the most likely to elicit a response, but these responses were infrequent (93% of cells had no response) (Figure 6A). All of the cells that responded to nerolidol had a sustained increase in calcium and did not respond further to the addition of capsaicin (seven of 96 cells) (Figure 6B). This, combined with capsaicin responses in cells that did not respond to nerolidol, suggests that nerolidol may have been acting on a target other than TRPV1. Among the remaining terpenoids, linalool (1 μM) infrequently (0.86%) elicited a brief increase in calcium (one of 116 cells) (Figure 6C,D).

According to Jansen and colleagues [16], myrcene activates TRPV1 at 5 μM. In our experiments, myrcene (1 μM) did not elicit a response in 98.7% of cells (Figure 6E). Similar to nerolidol, the cells (1.3%) that responded to myrcene did not further respond to capsaicin, indicating that myrcene may be acting on an infrequently expressed channel other than TRPV1 (two of 150 cells) (Figure 6F). Limonene and pinene did not elicit significant calcium responses in any DRGs tested (Figure 7A,B).

## 3. Discussion

Non-regulated and often pleasingly scented, terpenoids are being actively introduced to a growing range of consumer products to enhance aroma or flavor [42], often accompanied by claims of therapeutic or other desirable effects. Terpenoids are also finding themselves in the scientific spotlight alongside phytocannabinoids due to a hypothesized ‘entourage effect’ with THC (reviewed in [43]). Though terpenoids are often proposed to interact with endocannabinoid signaling, this has not been studied in a systematic way in native expression systems. We tested a panel of terpenoids frequently present in cannabis-- myrcene, linalool, limonene, α-pinene, and nerolidol—based on their range of claimed effects, using two neuronal models. We found that terpenoids had little if any effect on presumed TRP channels in DRG neurons, though nerolidol did activate a sustained calcium response in a small (7%) subpopulation of DRGs. In the autaptic neuronal model we found that most terpenoids had little or no effect on basal synaptic transmission or on cannabinoid signaling at 1μM. The notable exception was nerolidol, which reduced cannabinoid signaling, perhaps by acting postsynaptically to inhibit some aspect of endocannabinoid synthesis proximal to DAGLα or delivery (release and transport across the synapse).

Terpenoids are often promoted as having effects via the cannabinoid signaling system, although recent studies indicate such effects may not occur via CB1 receptors [6,7]. For some terpenoids the claims about their activity are more explicit than others. Myrcene in particular is commonly considered to enhance the cannabis high. Since the euphoric effects of cannabis are mediated by cannabinoid CB1 receptors myrcene’s enhancement often has been assumed to involve augmentation of CB1 signaling. Our results, as well as other studies [6,7], indicate that for many terpenoids, including myrcene, this is unlikely, though they do not rule out effects through other cannabinoid-associated receptors such as CB_2_ or GPR18 or non-cannabinoid receptor-associated pathways. Terpenoids are structurally diverse (Figure 1); we tested a limited panel of compounds, selected based on their range of hypothesized effects, but other terpenoids may be of interest, especially given that one terpenoid, nerolidol, as noted above, exhibited an interesting signaling profile. The suppression of DSE by nerolidol, without antagonizing the effect of bath-applied 2-AG, suggests nerolidol inhibits a component of 2-AG production or perhaps its transport out of dendrites or across the synaptic cleft. Activity of DAGLα and DAGLβ enzymes that synthesize 2-AG in autaptic neurons, were unaffected by nerolidol. Nerolidol, therefore, presumably acts at another step during the life of 2-AG and, as such, may represent an interesting tool to explore endocannabinoid signaling and function. Another possibility is that nerolidol accelerates the metabolism of 2-AG presynaptically. However, this would be expected to result in a sharp peak DSE inhibition followed by rapid decay, similar to the rapid DSI inhibition that we have described in inhibitory autaptic neurons where 2-AG is metabolized by a combination of MAGL and COX2 [12,44], but this was not observed.

As noted above, several studies (e.g., [16]) have now reported effects of terpenoids on TRP channels, a diverse family of ion channels. However, the concentrations at which effects are observed are generally in the mid-micromolar range or higher. We therefore also tested this panel of terpenoids in a neuronal model that natively expresses several TRP channels, most notably TRPV1, TRPA1 and TRPM8 [38,39]. In DRGs we did not see reliable calcium increases by terpenoids, though again nerolidol proved to be a partial exception in that nerolidol did elicit a sustained calcium response in a small subset (~7%) of DRGs. Nerolidol may therefore target a protein that allows entry of calcium from outside the cell or release from intracellular stores.

An important question in evaluating the effects of terpenoids is the likely physiological concentration of a given terpenoid that a person can realistically achieve in their body. As a starting concentration, 1 μM was chosen because it is at the upper end of physiological concentrations presumably to be encountered by consumers. In one study, goats (55 kg) were fed one gram each of limonene and α-pinene, yielding peak blood-plasma concentrations of 1 μM for limonene, and 600 nM for α-pinene [45]. In a separate study of plasma concentrations in a single human subject after topical application (i.e., massage) with lavender essential oil (1.5 g, containing 24.7% linalool), peak blood-plasma concentrations of linalool were found to be ~600 nM at 19 min after application, with a half-life of 13.8 min [46]. In principle, therefore, the substantial ingestion or topical application and systemic absorption of products containing high levels of a given terpenoid could result in concentrations approaching 1 μM, though this may vary from one terpenoid to another. For instance, the Poulopoulou study also examined levels of beta-caryophyllene after ingestion of 1 g, finding far lower net plasma concentrations than for α-pinene or limonene.

Though the complement of aromatic terpenoids largely determines the scent profile of a given cannabis strain, cannabis vendors have not hesitated to promote strains as having particularly healthful or otherwise desirable properties based on the dominant terpenoids present in that strain. However, it is difficult to see how cannabis users would see substantial levels of terpenoids after consumption of cannabis. Levels of a given terpenoid in cannabis are low (seldom exceeding 1%) and far lower than levels of THC (10–20% or more) [47]. The psychoactivity of THC therefore limits the amount of terpenoid that an individual is likely to consume. Taking the example of the high-myrcene Marionberry strain of cannabis (1.42%), if one were to incorporate 1 oz (28 g) of this strain into a brownie recipe, this would yield ~400 mg of myrcene. Accordingly, even eating the entire batch of brownies would not approach an intake of 1g of myrcene. A similar back-of-the-envelope calculation is not encouraging for those aiming for the mango high. Levels of myrcene in mango vary considerably (0.09–1.29 mg/kg) but assuming 1 mg/kg and an average mango pulp weight of 200 g, one arrives at well over a thousand mangoes to achieve anything approaching 1 g of myrcene. There may also be unexplored adverse effects connected with the ingestion of terpenoids [48].

Given the likely upper limits on terpenoid concentrations, it is important to note that many of the reported effects of terpenoids in vivo and in vitro are seen at very high concentrations. For example, it takes 100 mg/kg of myrcene or 50 mg/kg of limonene produce muscle relaxation in rats [49], while activation of TRPV1 channels required 150 µM [16]. Critical evaluation of some claims that factor in likely concentrations of these compounds do not thus far support roles of terpenoids acting at cannabinoid receptors. For example, Harris et al. showed that the analgesic properties of cannabis are mediated by THC, not terpenoids [50].

In the wake of the recent and continuing moves to legalize cannabis, terpenoids are finding their way into numerous products, often accompanied by claims of healthful or desirable effects. Some of the effects of terpenoids have been proposed to occur by altering the effects of cannabis. Our critical evaluation of a panel of these compounds in two neuronal models suggests that most terpenoids are unlikely to interact with CB_1_-based cannabinoid signaling or most DRG-expressed TRP channels at concentrations that are likely to be encountered by consumers. These results are in concordance with other recent studies of terpenoid action at CB1 receptors [7,51]. The notable exception—nerolidol—also did not alter CB1 signaling but instead inhibited endocannabinoid production through an undetermined mechanism, one that did not involve altering the activity of DAGLα or DAGLβ. The finding that a terpenoid alters endocannabinoid production through a novel mechanism is nonetheless a notable finding and raises the possibility that the remaining terpenoids may have additional surprises in store.

## 4. Materials and Methods

**Drugs.** All terpenoids (β-Myrcene, 7-Methyl-3-methylene-1, 6-octaciene; linalool, (±)-3,7-Dimethyl-3-hydroxy-1,6-octadiene; limonene, (R)-4-Isoproprenyl-1-methyl-1-cyclohexene; α-pinene, (1R, 5R)-2,6,6-Trimethylbicyclo[3.1.1]hept-2-ene; nerolidol, cis/trans mixture of 3-Hydroxy-3,7,11-trimethyl-1,6,10-dodecatriene) were obtained from Sigma-Aldrich (St. Louis, MO, USA). Drugs were racemic mixtures of enantiomers if stereochemistry is not indicated. 2-AG was purchased from Cayman Chemical (Ann Arbor, MI, USA).

**Hippocampal culture preparation.** All procedures used in this study were approved by the Animal Care Committee of Indiana University and conform to the Guidelines of the National Institutes of Health on the Care and Use of Animals. Mouse hippocampal neurons isolated from the CA1-CA3 region were cultured on microislands as described previously [52,53]. Neurons were obtained from animals (age postnatal day 0–2) and plated onto a feeder layer of hippocampal astrocytes that had been laid down previously [54]. Cultures were grown in high-glucose (20 mM) DMEM containing 10% horse serum, without mitotic inhibitors and used for recordings after 8 days in culture and for no more than three hours after removal from culture medium.

**Electrophysiology.** When a single neuron is grown on a small island of permissive substrate, it forms synapses—or “autapses”—onto itself. All experiments were performed on isolated autaptic neurons. Whole cell voltage-clamp recordings from autaptic neurons were carried out at room temperature using an Axopatch 200A amplifier (Molecular Devices, Sunnyvale, CA, USA). The extracellular solution contained (in mM) 119 NaCl, 5 KCl, 2.5 CaCl_2_, 1.5 MgCl_2_, 30 glucose, and 20 HEPES. Continuous flow of solution through the bath chamber (~2 mL/min) ensured rapid drug application and clearance. Drugs were typically prepared as stocks, and then diluted into extracellular solution at their final concentration and used on the same day.

Recording pipettes of 1.8-3 MΩ were filled with (in mM) 121.5 KGluconate, 17.5 KCl, 9 NaCl, 1 MgCl_2_, 10 HEPES, 0.2 EGTA, 2 MgATP, and 0.5 LiGTP. Access resistance and holding current were monitored and only cells with both stable access resistance and holding current were included for data analysis. Conventional stimulus protocol: the membrane potential was held at –70 mV and excitatory postsynaptic currents (EPSCs) were evoked every 20 s by triggering an unclamped action current with a 1.0 ms depolarizing step. The resultant evoked waveform consisted of a brief stimulus artifact and a large downward spike representing inward sodium currents, followed by the slower EPSC. The size of the recorded EPSCs was calculated by integrating the evoked current to yield a charge value (in pC). Calculating the charge value in this manner yields an indirect measure of the amount of neurotransmitter released while minimizing the effects of cable distortion on currents generated far from the site of the recording electrode (the soma). Data were acquired at a sampling rate of 5 kHz.

DSE stimuli: After establishing a 10–20 s 0.5 Hz baseline, DSE was evoked by depolarizing to 0 mV for 50 msec, 100 msec, 300 msec, 500 msec, 1 sec, 3 sec and 10 sec, followed in each case by resumption of a 0.5 Hz stimulus protocol for 20–80+ s, allowing EPSCs to recover to baseline values. This approach allowed us to determine the sensitivity of the synapses to DSE induction. To allow comparison, baseline values (prior to the DSE stimulus) are normalized to one. DSE inhibition values are presented as fractions of 1, i.e., a 50% inhibition from the baseline response is 0.50 ± standard error of the mean. The *x*-axis of DSE depolarization-response curves are log-scale seconds of the duration of the depolarization used to elicit DSE. Depolarization response curves are obtained to determine pharmacological properties of endogenous 2-AG signaling by depolarizing neurons for progressively longer durations (50 msec, 100 msec, 300 msec, 500 msec, 1 sec, 3 sec and 10 sec). The data were fitted with a nonlinear regression (Sigmoidal dose response; GraphPad Prism 7.03, La Jolla, CA, USA), allowing calculation of an ED_50_, the effective dose or duration of depolarization at which a 50% inhibition is achieved. A statistically significant difference between these curves is defined as non-overlapping 95% confidence intervals of the ED_50_s. Values on graphs are presented as mean ± S.E.M.

**Enzchek lipase activity assay.** Multiple 10 cm dishes of Chinese hamster ovary (CHO) cells were grown in parallel in Ham’s F12 medium. When cells reached ~70% confluence, one dish was transfected with DAGLα and mCherry plasmids using lipofectamine 2000 (Thermo Fisher Scientific, Waltham, MA, USA) following the manufacturer’s instructions. The lipase activity of protein extracts of these cells was then compared with untransfected CHO cells on the next day using an Enzchek lipase substrate assay (Thermo Fisher). Protein was extracted from cells by treating them with a freshly prepared lysis buffer consisting of 50 mM HEPES, 100 mM NaCl, 5 mM CaCl2, 0.5% TritonX-100, pH 7.2. A Pierce protease inhibitor mini tablet (Thermo Fisher) was added to 25 mL of Lysis buffer. Cells were treated with 1mL ice-cold lysis buffer for 5 min and centrifuged for 5 min at 4 °C after which the supernatant was taken and maintained on ice for experiments the same day.

Samples were tested on a 96 well plate using a Flexstation III (Molecular Devices, San Jose, CA, USA). Wells received protein extract (50 µL); lysis buffer (50 µL); 3 µM nerolidol or vehicle (5 µL). Enzchek (20 µM in 5 µL) was added just before readings were taken (485 nm stimulation/515 nm emission, six readings per time point, 30 sec intervals). Samples were run in pairs or triplicates (technical replicates) in a given experiment. Experiments were repeated from separate cultures (biological replicates). Slopes from the linear phase of the time courses (rise time per minute divided by micrograms of protein per sample) were calculated to estimate lipase activity. Values for a given experiment were normalized to their vehicle control for that day and compared using an unpaired Student’s *t*-test.

**Gel-based competitive ABPP experiments.** Mice were sacrificed, and brains removed, rinsed with lysis buffer (50 mM HEPES, 100 mM NaCl, 5 mM CaCl_2_, 0.5% TritonX-100, pH 7.2, protease inhibitor as above), and placed in Dounce homogenizer with 2mL of cold lysis buffer and homogenized. Homogenates were maintained on ice for 15 min, then transferred to Eppendorf tubes and spun down (1000× *g*, 5 min, 4 °C). The supernatant was collected, and protein concentration determined using Bradford’s assay. Samples were used the same day. For gel-based competitive ABPP experiments, proteomes (1 mg/mL) were treated with drug or vehicle for 30 min, then treated with HT-01 (1 μM final concentration) in a 50 μL total reaction volume. After 30 min at 37 °C, the reactions were quenched with SDS-PAGE loading buffer (In 10 mL: 2 mL 1M Tris-HCl, pH 6.8; 0.8g SDS; 4 mL 100% glycerol; 1 mL 0.5M EDTA; 0.2% bromophenol blue; 4% β-mercaptoethanol). After separation by SDS-PAGE (10% acrylamide), samples were visualized by in-gel fluorescence scanning using a Typhoon 9500 fluorescent scanner. Band intensity was quantified using Fiji. To confirm the identity of the HT-01 labeled DAGLβ band, we tested whether signal of the predicted 74 kDa target band was blocked by the DAGLβ blocker KT109 [55] (100 nM). Experiments were run from two separate samples and compared to the hypothetical non-effect (1.0) using a one-sample Student’s *t*-test.

**Dorsal root ganglion cell culture.** DRGs were harvested from P-0 through P-14 day-old rat pups following IACUC approval and following guidelines for the ethical care and use of laboratory animals. Rats were euthanized using isoflurane inhalation and cervical dislocation. DRGs were harvested using the protocol described by Sleigh et al. [56]. Briefly, rats were sprayed with 70% ethanol and the dorsal side opened along the longitudinal axis with surgical scissors. The spine was removed, cleaned of excess muscle, and cut longitudinally along the dorsal and ventral surfaces, then placed into cold Dissection Solution (Earl’s Balanced Salt Solution (Gibco, 24010043), 10 mM MgCl_2,_ 1X Glutamax (Gibco, 35050061), Penicillin/Streptomycin (500 ug/mL, Gibco, 15140122), and 10 mM HEPES (Fisher Scientific, BP310-1) and the spinal cord was carefully removed and discarded. DRGs were pulled from the vertebrate and placed into a 15 mL conical centrifuge tube containing ice-cold Dissection Solution. DRGs were centrifuged at 100× *g* at 4 °C and the media was replaced with Dissection Solution containing 10 mg/mL collagenase type II (Gibco, 17101015). DRGs were incubated at 37 °C for 20 min. Trypsin/EDTA (Gibco, 15090046) was then added to a final concentration of 0.05% and the tissue was incubated a further 3 min. Tissue was centrifuged at 4 °C as above and washed three times in ice-cold DMEM (Gibco, 11965126) containing 10% fetal calf serum. DRGs were then triturated 30 times in 4 mL of this medium, centrifuged, and resuspended in 3 mL of cold culture medium (Neurobasal A (Gibco, 10888022), 2.5 mg/mL insulin, 5 mg/mL transferrin, 5 mg/mL nerve growth factor-β (Sigma, SRP4304), 1X B27 (Gibco, 17504044), 1X Glutamax, 10% Fetal calf serum). Cells were then counted and plated at a concentration of ~5000 cells/cm^2^ on coverslips coated with poly-d lysine (Sigma, P-7886) and laminin (Millipore, SCR127, Burlington, MA, USA). Cells were cultured at 37 °C and 5% CO_2_, with one half of the media changed every 3–4 days.

**Calcium imaging.** DRGs were cultured for 2–10 days before the experiment. DRGs were treated with Fluo4-AM (5 µM) for 30 min at 37 °C, after which the cells were washed in extracellular solution (see electrophysiology) for 20 min to allow for de-esterification of Fluo4-AM. Fluorescence was then monitored on a Nikon TE200 inverted microscope with a 10× objective, a Hamamatsu Flash 4.0 camera and Nikon Elements AR software which controlled a SpectraX light engine for stimulation of fluorescence. Target DRGs were chosen before calcium imaging based on neuronal morphology using a brightfield image. This brightfield mask was then applied to the image series. Images were acquired every 30 s for 15 min and then analyzed using FIJI software with the 1-click ROI manager plugin [57], to measure the change in fluorescence intensity over 15 min. Baseline fluorescence intensity was normalized to zero based on the two minutes preceding drug application.

## Figures and Tables

**Figure 1 molecules-27-05655-f001:**
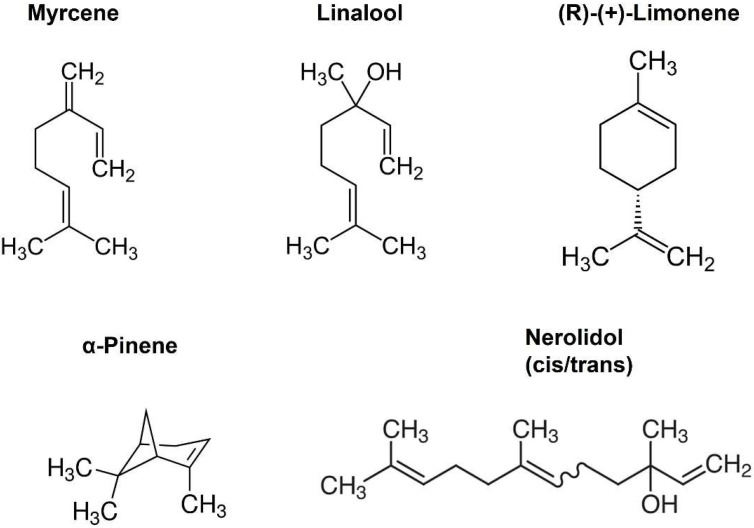
Structures of selected terpenoids.

**Figure 2 molecules-27-05655-f002:**
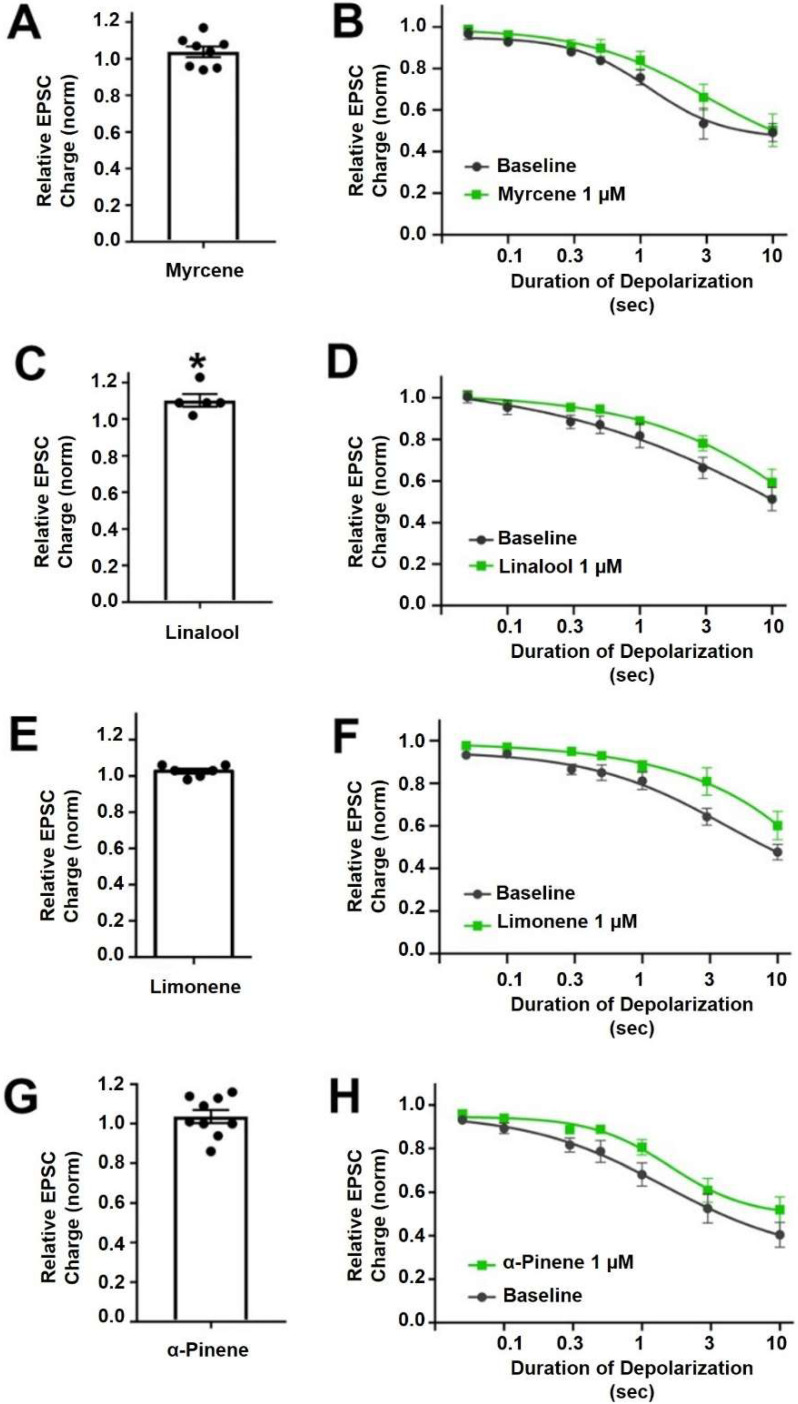
**Myrcene, linalool, limonene and****α-pinene do not alter cannabinoid signaling in autaptic hippocampal neurons.** (**A**) Myrcene (1 µM) did not alter EPSCs. (**B**) Myrcene also did not have an effect on DSE responses. (**C**) Linalool (1 µM) slightly increased EPSCs. (**D**) However, linalool had no effect on DSE responses. (**E**) Limonene (1 µM) did not alter EPSC sizes (**F**) and did not significantly inhibit DSE responses. (**G**) α-pinene (1 µM) did not alter EPSCs. (**H**) Similarly α-pinene did not inhibit DSE responses. * *p* < 0.05 one-sample *t*-test vs. 1.0 or vs. baseline at 3 s of depolarization. Mean ± SEM shown.

**Figure 3 molecules-27-05655-f003:**
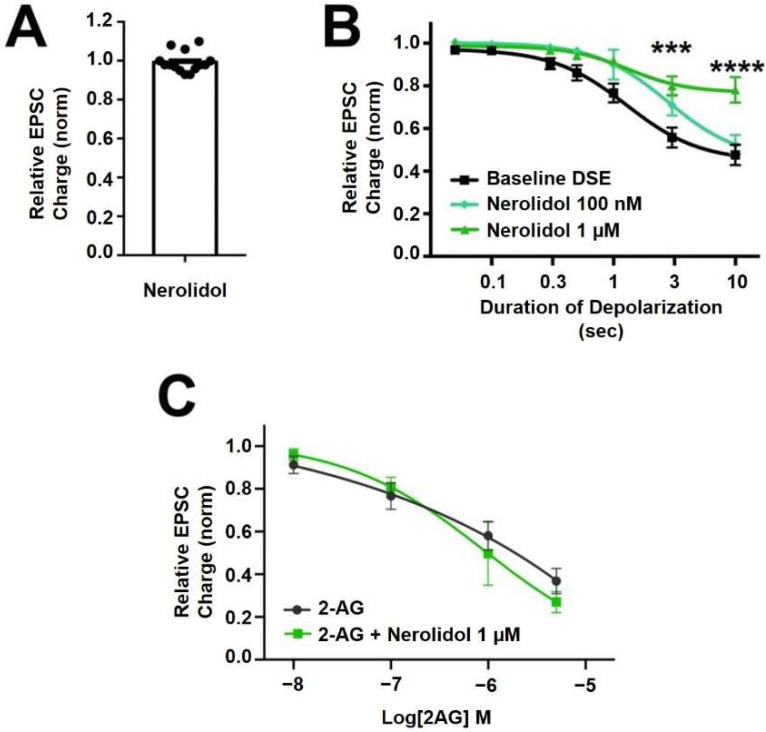
**Nerolidol has complex interaction with cannabinoid signaling.** (**A**) Nerolidol (1 μM) does not alter EPSC sizes. (**B**) Nerolidol substantially inhibits DSE responses at 1 μM but not at 100 nM. (**C**) 2-AG signaling is not diminished by treatment with nerolidol. *** *p* < 0.001, **** *p* < 0.0001 at 3 and 10 s depolarizations via 2-way ANOVA with Bonferroni post hoc test.

**Figure 4 molecules-27-05655-f004:**
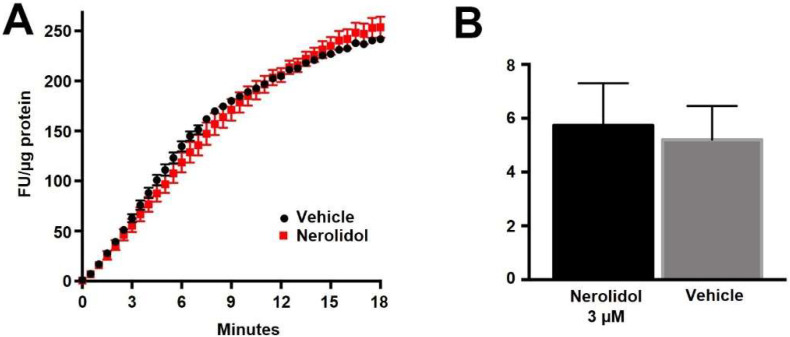
**Nerolidol does not alter the basal activity of DAGLα.** (**A**) Sample Enzchek lipase activity time course in the presence of vehicle (black) or nerolidol (3 μM, red) in DAGLα-transfected CHO cells. (**B**) Summarized results show lipase activity in DAGLα-expressing CHO cells in response to vehicle or nerolidol (*n* = 3).

**Figure 5 molecules-27-05655-f005:**
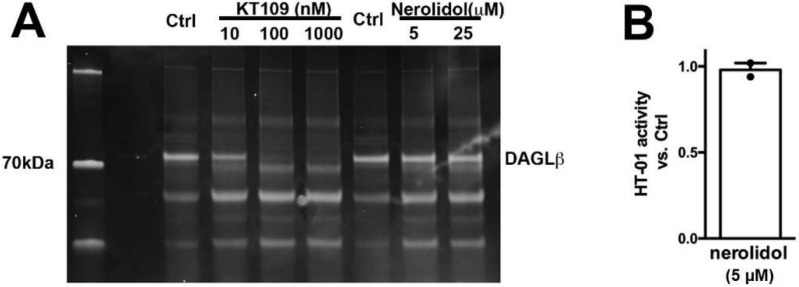
**Nerolidol does not compete for binding at the active site of DAGLβ.** (**A**) DAGL-binding probe HT-01 is competed by DAGLβ blocker KT109 (100 nM) but not by nerolidol. (**B**) Summary of results from 2 experiments. NS by one tailed *t*-test vs. 1.0 (1.0 = no effect).

**Figure 6 molecules-27-05655-f006:**
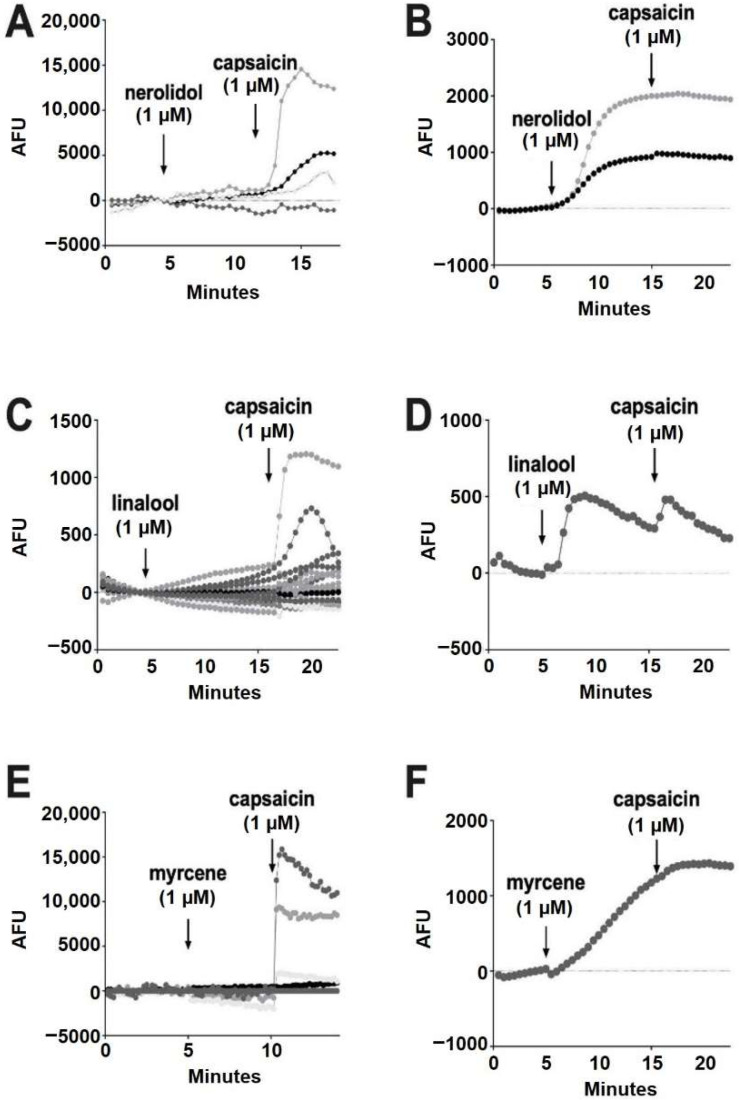
**Nerolidol, linalool and myrcene do not reliably activate calcium responses in DRGs.** (**A**) Sample population response using a calcium sensor in DRGs treated with nerolidol (1 μM) followed by capsaicin (1 μM). (**B**) Infrequently (7% of neurons), nerolidol elicited a sustained calcium response as shown in these sample time courses. (**C**) Time course showing responses to linalool (1 μM) as in A. (**D**) Rarely (0.86%) linalool elicited a response in DRGs. (**E**) Time course showing responses to myrcene (1 μM) as in A. (**F**) Myrcene rarely (1.3%) elicited a calcium response in DRGs.

**Figure 7 molecules-27-05655-f007:**
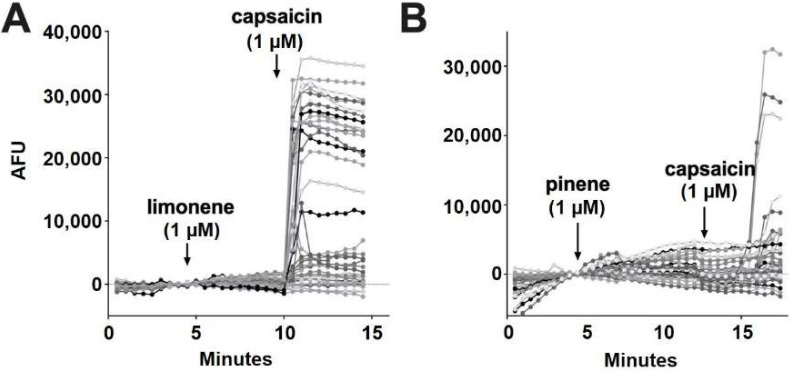
**Limonene and pinene did not activate calcium responses in DRGs.** (**A**) Sample population response using a calcium sensor in DRGs treated with limonene (1 μM) followed by capsaicin (1 μM). (**B**) Population response to pinene (1 μM) treatment as in A.

**Table 1 molecules-27-05655-t001:** Effective dose 50 (ED50, with 95% confidence interval) calculated from depolarization-response curves indicates the duration of depolarization (in sec) necessary to produce 50% maximal response. Maximal inhibition shows relative EPSC values at the longest depolarization (10 s, 1.0 = no inhibition).

	Terpenoid			Control		
	ED50 (sec)	95% CI	Max. Inhibition	ED50 (sec)	95% CI	Max. Inhibition
Myrcene	3.50	0.3–32.2	0.50	1.14	0.67–5.19	0.49
Linalool	31.31	ambiguous	0.59	22.1	ambiguous	0.51
Limonene	8.59	1.6–46.1	0.60	3.24	1.5–7.1	0.48
α-pinene	1.60	0.88-2.88	0.79	1.45	0.36-5.82	0.58
Nerolidol (100 nM)	3.33	ambiguous	0.64	2.28	1.15–110.7	0.56
Nerolidol (1 μM)	6.97	ambiguous	0.81	1.32	0.68–23.22	0.49

## Data Availability

The data underlying this article will be shared on request to the corresponding author.

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
