# Peer review of "A Critical Evaluation of Terpenoid Signaling at Cannabinoid CB1 Receptors in a Neuronal Model"

_molecules, 2022, doi:10.3390/molecules27175655_

Round 1

Reviewer 1 Report

The authors used two neuronal models (autaptic hippocampal neurons and DRG neurons) to evaluate the efficacy of five cannabis terpenoids on neuronal cannabinoid signaling. The study was well-designed and the manuscript was well-written. Some minor points need to be addressed:

CB1, 2-AG: full name needs to be stated when it first appears.

Lind 35: please provide some examples.

I suggest adding introduction of the neural models you used in this study.

Did you add protease inhibitor in the process of protein extraction? At what concentration?

There are several unnecessary spaces throughout the manuscript. 

Author Response

We thank the reviewer for their close reading of our manuscript and for the constructive suggestions.  We have listed the reviewer comments in bullet form below with our reply below each point.

  • CB1, 2-AG: full name needs to be stated when it first appears.
    • We have now added this to the manuscript.
  • Line 35: please provide some examples.
    • We have now added several examples here.
  • I suggest adding introduction of the neural models you used in this study.
    • We have now added information about the neuronal models to the introduction (starting at line 52).
  • Did you add protease inhibitor in the process of protein extraction? At what concentration?
    • Yes, thank you for pointing this out. We did and have added that information to the methods section (lines 395 and 411).
  • There are several unnecessary spaces throughout the manuscript. 
    • We have reviewed the manuscript for layout issues.

Reviewer 2 Report

This manuscript seems to be a continuation of authors' previous study on understudied phytocannabinoids (included as Ref. 9) in the manuscript and this time, they choose five terepene derivatives for similar study. It presents interesting results however, I have some questions regaridng the compounds.

1) Linalool also has two enantiomers [(S)-(+)-linalool  and (R)-(–)-linalool] and authors have not mentioned which enantiomer (or racemic mixture) they used. They used a single enantiomer of limonene but it is not clear for linalool. As enantiomers usually show different biological activity, it should be clarified.

2) Stereochemistry of a-pinene and nerolidol is also not clear in structures (figure 1).

3) Although in mathodology, this manuscript has high similarity with authors' previous manuscript (ref. 9)

Author Response

We thank the reviewer for their close reading of our manuscript and for the constructive suggestions.  We have listed the reviewer comments in bullet form below with our reply below each point.

  • Linalool also has two enantiomers [(S)-(+)-linalool  and (R)-(–)-linalool] and authors have not mentioned which enantiomer (or racemic mixture) they used. They used a single enantiomer of limonene but it is not clear for linalool. As enantiomers usually show different biological activity, it should be clarified.
    • Yes this point is well-taken. We have updated the manuscript accordingly (starting at line 334). Unless otherwise indicated, the drugs used were racemic mixtures of enantiomers.
  • Stereochemistry of a-pinene and nerolidol is also not clear in structures (figure 1).
    • We have updated this figure accordingly.
  • Although in methodology, this manuscript has high similarity with authors' previous manuscript (ref. 9).
    • This is correct, as noted by Reviewer 1 this manuscript is a companion to our study of a panel of phytocannabinoids published last year, using many of the same methodological approaches. The methods are therefore similar.